# Effect of Environmental Enrichment on the Brain and on Learning and Cognition by Animals

**DOI:** 10.3390/ani11040973

**Published:** 2021-03-31

**Authors:** Thomas R. Zentall

**Affiliations:** Department of Psychology, University of Kentucky, Lexington, KY 40506-0044, USA; zentall@uky.edu

**Keywords:** enrichment, isolation, learning, impulsivity

## Abstract

**Simple Summary:**

Most people consider the environment in which animals are kept to be an ethical matter, separate from the research that we conduct with them. Those of us who do research on the cognitive behavior of animals try to consider their welfare, but what we often fail to recognize is that the welfare of the animals we study can affect the results of experiments that we investigate. We have but scratched the surface of the question, how do enriched environments affect the cognitive behavior of animals, in our case pigeons. We have found that pigeons with experience in an enriched environment are less impulsive. The reduction in impulsivity results in a reduced tendency to make the suboptimal choice. It also has been claimed to make animals more optimistic, as assessed by their tendency to make choices of more favorable alternatives, under ambiguous conditions.

**Abstract:**

The humane treatment of animals suggests that they should be housed in an environment that is rich in stimulation and allows for varied activities. However, even if one’s main concern is an accurate assessment of their learning and cognitive abilities, housing them in an enriched environment can have an important effect on the assessment of those abilities. Research has found that the development of the brain of animals is significantly affected by the environment in which they live. Not surprisingly, their ability to learn both simple and complex tasks is affected by even modest time spent in an enriched environment. In particular, animals that are housed in an enriched environment are less impulsive and make more optimal choices than animals housed in isolation. Even the way that they judge the passage of time is affected by their housing conditions. Some researchers have even suggested that exposing animals to an enriched environment can make them more “optimistic” in how they treat ambiguous stimuli. Whether that behavioral effect reflects the subtlety of differences in optimism/pessimism or something simpler, like differences in motivation, incentive, discriminability, or neophobia, it is clear that the conditions of housing can have an important effect on the learning and cognition of animals.

## 1. The Effect of Environmental Enrichment on the Brain and Learning of Animals

There has been increasing research interest in improving the welfare of captive animals by exposing them to enriched environments. This has been done primarily for humane reasons; however, there is a long history of research on how environmental enrichment affects the brain and behavior of animals. These effects suggest that environmental enrichment should be explored for other than purely humane reasons. The purpose of this article is to identify several of the effects of enriched environments on brain and behavior, some of which may have important implications for cognitive functioning and even for treating disordered human behavior.

Interest in the effects of environmental enrichment can be traced as far back as Darwin [1]. In accounting for the difference in the size of the brain of domestic versus wild rabbits, Darwin proposed that the added environmental enrichment of wild rabbits might be responsible. At a behavioral level, Hebb [2] hypothesized that animals raised in enriched environments may demonstrate enhanced problem-solving capabilities. His conjecture was based on the finding that rats raised as pets were more adept at maze learning than laboratory rats. 

When environmental enrichment is studied in the laboratory, although the protocols have varied, they generally involve time spent in a large cage with novel objects and several conspecifics (compared with a more typical smaller cage, with single or double animal housing). 

## 2. Environmental Enrichment and the Brain

Pioneering research on the effect of environmental enrichment on the brain was conducted by University of California, Berkeley investigators in the 1960s and 1970s (See [3], for a summary). For example, Krech, Rosenzweig and Bennett [4] found that environmental enrichment resulted in an increase in the weight of the visual and somatosensory cortex in rats in comparison to rats kept in isolation (see also [5,6]). More specifically, Diamond et al. [7] found that rats exposed to environmental enrichment had increased cortical thickness, especially in the occipital cortex. Environmental enrichment has also been shown to produce changes in the auditory cortex [8].

At the level of neurons, environmental enrichment has been found to result in increases in the size of neuronal cell bodies and nuclei, the number and size of dendrites, as well as increasing dendritic branching and the number of dendritic spines [9,10]. Environmental enrichment also has been found to produce alterations in glial cells in the brain [11,12].

At a gross level, following a brain injury during infancy, environmental enrichment has been found to aid in the repair of the brain [13]. Specifically, following environmental stimulation, the density of microglia associated with a brain injury has been found to decrease, as indicated by immunohistochemical staining [13]. 

Environmental enrichment also has been found to increase the number of blood capillaries in the brain, as well as an increase in metabolic activity, indicated by an increase in the number of mitochondria [14]. Research has also found that environmental enrichment has altered neuronal function in the medial prefrontal cortex, an area of the brain that has been implicated in the reinforcing efficacy of drugs of abuse and in spatial working memory [15].

Social rearing alone appears to play an important role in some of these brain effects. For example, rats raised socially even in the absence of novel objects or larger cages have increased monoamine neurotransmitter levels in mesocorticolimbic structures [16].

## 3. Environmental Enrichment and Drugs of Abuse

Many studies have shown that environmental enrichment alters the neurochemical effects of various drugs of abuse. For example, rats exposed to an enriched environment have greater extracellular levels of dopamine in the nucleus accumbens following intravenous introduction of amphetamine, compared to normally housed rats [17]. It has been hypothesized that repeated novelty exposure can sensitize limbic structures, resulting in greater dopamine release [17]. In addition to increases in dopamine, enrichment alters drug-induced glutamate release, as measured by microdialysis [18], and rats given environmental enrichment are less sensitive to the reinforcing effect of amphetamine, especially at low doses [19].

Furthermore, rats raised with environmental enrichment are less sensitive to both the acute and repeated stimulant effects of nicotine [20], and Deehan et al. [21] found that operant responding for oral doses of alcohol was significantly lower in environmentally enriched rats compared to isolated rats. Isolated rats often show a preference for alcohol over water, while enriched rats do not. Thus, considerable research suggests that environmental enrichment protects animals against drug abuse vulnerability. 

One of the behavioral mechanisms that may be responsible for lessened susceptibility to drugs of abuse by rats is the effect that enrichment appears to have on impulsivity [22,23]. For example, enriched rats show less impulsivity (as measured by the slope of delayed discounting functions) in the acquisition of tasks involving conditioned reinforcement [24]. Thus, rats that live in an enriched environment appear to show a lower level of drug self-administration because they have greater inhibitory control.

Interestingly, similar effects have been found in humans. Enriching environments can serve as protective factors that can decrease drug use among addiction-vulnerable adolescents and adults [25].

Drug abuse has been viewed by some as a pathology of decision making [26]. In the context of decision making, one is always faced with decisions among activities that have different values. Under typical housing conditions, the number of behavioral activities is generally quite limited. When animals have been exposed to enriched environments, however, the number of possible activities they can engage in is greatly expanded. This idea, which derives from behavioral economics [27,28], proposes that for animals exposed to an enriched environment, the relative value of drugs is smaller than the relative value of the other activities otherwise available to them. This behavioral account suggests that the changes in behavior may not require changes in brain structures associated with long-term exposure to enriched environments. 

## 4. Enrichment as a Treatment for Attention Deficit Hyperactivity Disorder 

Children with attention deficit hyperactivity disorder are characterized by their impulsivity and their inability to focus on tasks [29]. Traditional theories of attention deficit hyperactivity disorder have hypothesized that these children are over-aroused and are best treated by reducing the amount of stimulation in their environment [30,31].

A more homeostatic model has been proposed that assumes that under normal conditions, such children, rather than suffering from overstimulation, are actually under-stimulated, relative to more typical children [32]. According to this theory, the distractibility of these children actually results from their attempt to increase stimulation from other sources, and by so doing, they tend to be distractable. Consistent with this hypothesis, there is evidence that environmental enrichment actually increases inhibitory control for these children [33]. It does so presumably by providing alternative sources of environmental stimulation, thus allowing them to focus better on the task at hand. This research supports the general conclusion that environmental enrichment can reduce impulsive behavior and allow for better attention to the requirements of a task, not only for children with attention deficit hyperactivity disorder, but also for more typical children.

## 5. Enrichment and Social Interaction 

Returning to the behavioral effects of environmental enrichment on animals, rats raised in enriched environments with other rats tend to be less aggressive [34] and are more skilled at providing aggression inhibiting cues to other rats [35]. This may result from the fact that socially housed rats are likely to have worked out dominance hierarchies, and learned what cues work to reduce the aggressive behavior of others. Whether environmental enrichment by itself, in the absence of other rats, would result in similar effects is not known. 

## 6. Enrichment and Learning

Enrichment of the environment has long been proposed as a treatment or strategy for increasing cognitive ability in rodents [36,37]. For example, Hullinger, O’Riordan and Burger [38] found that rats that had received 1 month of environmental enrichment showed better learning of the Morris Water Maze (as measured by a greater number of goal crossings on test trials in the absence of the training platform), and more exploration of a novel test object, than normally housed rats. Some studies have suggested that the improvement in Morris Water Maze learning is due to rapid acquisition and the flexible use of spatial information [39], while others suggest that environmental enrichment may have a greater impact on the processes of consolidation, having found that enriched animals show better maintenance of spatial information shortly after completing training [40].

A positive effect of housing in an enriched environment has also been found in visual discrimination learning [41]. Interestingly, given that impulsivity is the mechanism proposed to be responsible for the lessened susceptibility to drug abuse by rats, Ough, Beatty, and Khalili [42] found superiority among rats raised in an enriched environment on schedules involving learned inhibition (a differential reinforcement of low rate of responding schedule). Similarly, there is evidence that the rats learn the passive avoidance of a candle flame faster than the more typically housed rats [43].

Enriched rats also appear to learn spatial tasks faster than isolated rats [44,45]. This may be due to the fact that they are more exploratory than controls, but enriched rats are also more reliant on extra-maze cues, as indicated by the greater disruption in their performance when the training maze is rotated. 

The effects of environmental enrichment on the rats’ learning are complicated by their different responses to novelty and their sensitivity to reinforcement and punishment. For example, the effects of enrichment on spatial learning are greater under conditions of high drive (24–36 h food deprivation) than low drive (12–14 h food deprivation) [46]. Under low drive, the isolated rats appear to show greater exploratory behavior than the enriched rats, whereas under high drive both groups show reduced exploratory behavior. Similarly, Renner and Rosenzweig [3] reported that isolated rats find some levels of shock more aversive than enriched rats, thus it may be difficult to equate the consequences of learning for rats exposed to different environments. That is, it may be difficult to separate performance differences, resulting from differences in motivation, from learning differences.

Dell and Rose [47] identified another difference in spatial learning resulting from differences in the housing environment of rats. In a spatial learning task, if one considers only initial errors, enriched rats do not learn faster than isolated rats. If one considers repeat errors, however, enriched rats tend to make many fewer total errors than isolated rats. This finding suggests, once again, that enriched rats learn to inhibit incorrect responding faster than isolated rats.

The brain mechanisms responsible for the facilitation of learning are not clear, but the changes in brain structure that have been found with exposure to an enriched environment [3] are likely to be involved.

## 7. Environmental Enrichment and Complex Behavior

For a variety of reasons, very few studies have examined the relation between enrichment and complex learning. Complex behavior can be defined as behavior not easily explained by simple associative (S-R) learning together with primary stimulus generalization. It may be that many investigators do not think that enriched environments would have such far ranging effects, but complex learning usually involves simpler elements, and those simpler elements may affect the way animals approach complex learning. In what follows are two examples of the effect of environmental enrichment on complex learning that can be traced to a simpler underlying process, namely reduced impulsivity.

## 8. The Suboptimal Choice Task 

Animals sometimes make choices that are considered suboptimal. For example, rats or pigeons may choose an alternative that gives them a smaller amount of food sooner over a different alternative that gives them a larger amount of food later (delay discounting) [48]. In fact, the slope of the delay discounting function (the reduced preference for the larger amount later as a function of its delay) has been used as a measure of the impulsivity of an animal [48]. Although we humans tend to view impulsivity as a negative attribute, depending on the species, some degree of impulsivity may actually be adaptive because in nature, delayed reinforcement often suffers a lower probability of reinforcement due to competition from others. 

Even if one controls for delay of reinforcement, animals sometimes make systematic choices that result in less reinforcement. In an analogy to human gambling behavior, pigeons show a preference for an alternative that occasionally (20% of the time) signals that they will receive a high-value reinforcer (10 pellets of food, the “jackpot”) but usually (80% of the time) signals that they will receive nothing, over an alternative that always signals that they will receive three pellets of food [49]. Curiously, research has shown that under these conditions, the signal for the absence of reinforcement does not acquire inhibition (as assessed by a combined cue test) [50]. Furthermore, in addition to the absence of inhibition associated with the signal for the absence of reinforcement, the signal for the high-value reinforcer, the “jackpot,” acquires more value than it should. It appears that there is positive contrast between the suboptimal alternative at the time of the choice and the signal for the high value reinforcer [51]. The positive contrast appears to result in an impulsive choice of the suboptimal alternative, as indicated by the positive correlation between the suboptimal choice and the slope of the delay discounting function [52]. 

The high degree of impulsivity in this suboptimal choice task encouraged us to ask if environmental enrichment might have an effect on the suboptimal choice. The environmental enrichment consisted of placement of pigeons in a large cage (2.44 m high, 1.23 m wide, and 2.44 m deep) together with three other pigeons. In the enrichment cage they also had access to shelves on which they could perch, a large pan with water, a large pan with sand and several hanging ornaments. 

Placement in the enrichment cage occurred for 4 h, shortly after each pigeon had completed its experimental session on the suboptimal choice task. When the pigeons were not in the enrichment cage or the experimental apparatus, they were in their home cage (a standard cage 28 cm wide, 38 cm deep, and 30.5 cm high). Control pigeons were treated the same, except they spent no time in the enrichment cage.

Pigeons were exposed to a version of the suboptimal choice task involving choice between an alternative that 50% of the time signaled 100% reinforcement, and 50% of the time signaled no food (the suboptimal alternative), and an alternative that 100% of the time signaled 75% reinforcement (the optimal alternative). The control group quickly showed a strong preference for the suboptimal alternative, whereas the environmentally enriched group chose optimally for many training sessions, before eventually choosing suboptimal alternative [53]. Even though the environmental enrichment took place following each experimental session for a relatively short time and only during the course of the experiment, this relatively small amount of enrichment had a significant effect on delaying the pigeons’ suboptimal choice.

The research on the effects of environmental enrichment on suboptimal choice by pigeons has implications for the treatment of human addictive gambling behavior. Providing addicted humans with a more enriched environment (e.g., outdoor activities) may provide them with behavioral alternatives to gambling, or even drugs of abuse.

## 9. Time Judgements

Judgements of the passage of time can be assessed using the peak procedure. In the peak procedure, animals are trained on a fixed interval schedule of reinforcement in which reinforcement follows the first response after a fixed duration. If after training on such a schedule, empty intervals are presented in which no reinforcement is provided, the rate of pecking, as a function of the time since the start of the trial, provides a measure of the animal’s subjective sense of the passage of time. Pecking typically increases from the start of the trial, reaches a peak at approximately the time reinforcement has occurred in the past, and then decreases.

An alternative procedure used to assess animal timing involves a temporal discrimination in which animals must discriminate between two stimulus durations. After experiencing stimulus duration A (e.g., 2 s), choice of comparison stimulus X is reinforced. After experiencing stimulus duration B (e.g., 8 s), choice of comparison stimulus Y is reinforced. Following training with such a temporal discrimination, one can present the animal with stimulus durations between the two training values to determine the subjective scale of the passage of time. Typically, the duration at which the animal chooses equally between the two comparison stimuli, “short” and “long” (the point of subjective equality), falls close to the geometric mean of the training durations (as would be expected from Weber’s law). 

Several factors can affect the psychophysical timing functions. For example, the timing function can be affected by what the animal is doing while timing. Specifically, there is evidence that if pigeons are required to peck at the timing stimulus, they tend to respond as if less time has passed than when they are required to refrain from pecking the timing stimulus [54]. Even when pecking is allowed but not required, they tend to judge that less time has passed than when they are required to refrain from pecking the timing stimulus. Required pecking may serve as a modest source of enrichment.

## 10. Enrichment and Timing

Spending time in an enriched environment can also affect pigeons’ judgement of the passage of time. Pigeons that have spent some time in an enriched environment tend to judge that less time has passed than pigeons that were not given an enrichment experience [55]. What is the relation between the effects of housing in an enriched environment and pecking? One can consider time in an enriched environment and pecking as both involving additional activities. Although (in the case of the enriched environment experience) the activities occur outside of the experimental context, they appear to have an effect similar to required pecking during the stimulus to be timed. In both cases, the activities appear to distract from attention to timing cues, thus underestimating the passage of time. 

The effect that environmental enrichment has on the passage of time and on reduced suboptimal choice may be similar. It would be interesting to test more directly the hypothesis that both of these result from the same underlying mechanisms by asking if environmental enrichment can reduce the slope of the delay discounting function, thus allowing an animal to wait longer for the larger, later reinforcer rather than choose the smaller, sooner reinforcer.

## 11. Environmental Enrichment and Cognitive Bias

Eysenck et al. [56] proposed that anxious humans will tend to make more negative (i.e., pessimistic) interpretations of ambiguous stimuli. Harding et al. [57] attempted to apply this concept to animals. They trained rats to press a lever for food in the presence of one tone but not in the presence of another and then tested them with tones in between. To induce “anxiety,” they exposed some rats to “unpredictable” housing (changing their housing often). Harding et al. found that the rats that were exposed to unpredictable housing showed longer latencies to respond to tones that were close to the training tone associated with food, than normally housed rats. The authors suggested that the unpredictable housing resulted in the reduced anticipation of a positive event; that is, it put the animals in a negative emotional state [58]. 

Other research compared rats housed in an enriched environment with those housed in standard cages [59]. The rats were first trained on a spatial discrimination and were then tested on intermediate spatial locations. The researchers found that the rats in standard housing showed a longer latency to approach the location closest to the location that in training was associated with the absence of food. The results were interpreted as suggesting that the unenriched rats displayed “less optimistic-like” judgements of an ambiguous location. 

Related research with European starlings trained the birds on a temporal discrimination (2 vs. 10 s), with a differential outcome (immediate food vs. delayed food) associated with each duration [60]. When they tested the birds with durations between 2 and 10 s, they found that birds that had experienced enriched housing were biased towards the immediate food outcome compared to isolated housing. The authors concluded that the birds that had experienced enriched housing had developed an “optimistic response bias.” The bias towards immediate food would appear to suggest increased impulsivity, however, given the ambiguity of the test stimuli, impulsivity was not likely involved. 

Although the effect of an enriched environment on latency and choice has been attributed to differences in “optimism” and “pessimism,” Mendl et al. [61] note that the differences in these response measures, as a function of different environments, may depend on several more parsimonious factors.

For example, when animals exposed to an enriched environment show a greater tendency to respond to an ambiguous cue that is similar to a negative training stimulus (one associated with a mild shock or the absence of food), it may be because those animals have learned to be less cautious (or more exploratory) because of their more varied housing experiences. Similarly, in the case of a shorter latency to an ambiguous cue similar to the positive training cue, animals exposed to an enriched environment may have learned to be less neophobic. Whether neophobia and pessimism are different concepts is not clear.

In studies involving a go/no-go response (e.g., Harding et al. [53]), environmental enrichment could affect the animal’s motivation, making it more active and thus more likely to make a go response. Such a pattern would give the animal the appearance of being more optimistic.

According to Mendl et al. [61], animals that have experienced a relatively deprived environment may exhibit a greater “attentional bias” than animals from a more enriched environment. That is, the deprived environment may cause the animals to generalize to a greater extent from a negative stimulus alternative to similar ambiguous stimuli. 

Mendl et al. [61] also suggest that the incentive value of the ambiguous stimuli may be affected by the animals’ housing condition. The incentive value of a positive outcome may be enhanced or conversely the negative value of a negative outcome may be reduced by exposure to an enriched environment.

As proposed by Eysenck [56], anxiety may affect one’s response to novelty, and an animals’ housing condition may affect its level of anxiety, especially when exposed to novel stimuli. Exposure to environmental isolation may make an animal react to an ambiguous stimulus with greater anxiety than exposure to an enriched environment. Presentation of ambiguous stimuli may also involve greater risk (uncertainty). As an enriched environment inherently exposes animals to a more variable environment, housing in such an environment may make animals more accepting of risk. Finally, exposure to an enriched environment may give animals experience choosing among alternative actions, and such experience may make it more likely that they would respond more readily to ambiguous stimuli. 

Whether the effects of housing conditions can be interpreted as inducing emotions of optimism or pessimism in animals is arguable because several alternative motivational and perceptual mechanisms may be involved that do not require the attribution of such subtle and subjective emotional states. It is clear, however, that housing conditions do affect the behavior of animals after they have learned a discrimination and they are exposed to novel stimuli. 

## 12. Conclusions

Environmental enrichment can be thought of as being important for the welfare of animals. Research has found, however, that environmental enrichment can have important effects on various neuroanatomical measures of the brain—in particular, on various physiological measures of the cortex. Not surprisingly then, environmental enrichment has been found to facilitate various kinds of learning. A review of the various effects of environmental enrichment on learning suggests that the mechanism by which environmental enrichment affects learning may be by reducing impulsive choices. There are two implications of these findings. First, if measures of learning in different species are to be meaningfully interpreted, the nature of the environment in which the animals are housed (or have access to) may have an important effect on the learning measures that are obtained. Enriched housing can also affect the way in which animals respond to novel stimuli (a cognitive bias). Second, as noted in the section on attention deficit hyperactivity disorder, environmental enrichment may have important effects on human learning and other behavior. There is growing evidence that even for typical children the degree to which they are exposed to a stimulating environment affects their learning ability [32]. Thus, for the humane treatment of animals it is certainly important to consider environmental enrichment, but the importance of an enriched environment extends well beyond their humane care to measures of learning, cognition, and their reaction to stimulus novelty.

## Data Availability

Data can be obtained from author at zentall@uky.edu.

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
