# Peer review of "Effect of Environmental Enrichment on the Brain and on Learning and Cognition by Animals"

_animals, 2021, doi:10.3390/ani11040973_

Round 1
Reviewer 1 Report
This review was well-written, succint but informative.
Minor comments:
Abtract line 4: period missing
Page 2, last sentence of first paragraph: awkward sentence, consider re-phrasing or breaking up into two
Page 2, 7th paragraph: remove comma after "Although"
Page 2, 8th paragraph: how is impulsivity/inhibitory control defined/characterized here?
Page 2, last paragraph: is this explanation an alternative to explanations based on changes in neural structure/function described earlier? That is, are reductions in impulsive behavior due to the ability to engage in alternative behaviors independent of reductions in impulsivity caused by neural changes induced by enrichment?
Page 3, social interaction section: are the results reported here specific to enrichment involving social enrichment (as opposed to housing complexity)?
Page 4, first paragraph: What is meant by high/low drive?
Page 4, second to last paragraph, sentence beginning with "Curiously": should this say require not acquire?
Page 6, starling example: does this bias towards immediate food outcome imply an impulsive choice?
Page 6, second to last paragraph: To me, "pessimism" (ie, expectation of negative outcomes) and neophobia (fear of the unknown) seem to be the same thing
Author Response
Responses appear below comments preceded by an *
Abtract line 4: period missing
*Fixed
Page 2, last sentence of first paragraph: awkward sentence, consider re-phrasing or breaking up into two
*made into 2 sentences
Page 2, 7th paragraph: remove comma after "Although"
*done
Page 2, 8th paragraph: how is impulsivity/inhibitory control defined/characterized here?
*definition provided
Page 2, last paragraph: is this explanation an alternative to explanations based on changes in neural structure/function described earlier? That is, are reductions in impulsive behavior due to the ability to engage in alternative behaviors independent of reductions in impulsivity caused by neural changes induced by enrichment?
*it is an alternative. Now clarified
Page 3, social interaction section: are the results reported here specific to enrichment involving social enrichment (as opposed to housing complexity)?
*Effects have not be separated
Page 4, first paragraph: What is meant by high/low drive?
*now defined
Page 4, second to last paragraph, sentence beginning with "Curiously": should this say require not acquire?
*no, acquired (now defined)
Page 6, starling example: does this bias towards immediate food outcome imply an impulsive choice?
*not really, now explained
Page 6, second to last paragraph: To me, "pessimism" (ie, expectation of negative outcomes) and neophobia (fear of the unknown) seem to be the same thing
*good point. Could be similar.
Reviewer 2 Report
Thank you for inviting me for reviewing this paper - It was an enjoyable read! The structure of the review is neat. The author shows the support for an enriched environment for animals. The view of the author is supported by studies that have shown the effects of enrichment on brain and behaviour. It covers a wide range of investigation topics. I entirely agree with the author that voices are increasing toward providing better welfare for animals. This review comes in timely and it will be a great start for the community to push more investigations forward, and hopefully come up a standard and basic protocol for keeping animals in the lab.
I only have a couple of minor comments, mostly about clarity (for people who are new to animal welfare, psychology etc but interested in these fields).
The author has only put forward the investigations in enrichment and complex behaviour (timing). But indeed, there are many sections missing the linkage between how enrichment affects learning performance, decrease impulsivity and increase attention for humans who have ADHD etc, would it be worth to have a section to talk about this relationship (i.e., a relationship between enrichment, neurochemical and learning)? Alternatively, the author could stress on the aim is to show the outcome (and not the 'how' specifically) of enrichment has on the brain and behaviour? I understand there is word limit here, but adding either of this will help to minimise queries (about the 'how) and focus on the main aim of the review.
Currently, most labs have their own way of giving enrichment to their animals, so as the author mentioned at the beginning that protocol in studies about enrichment varied. If the author strongly supports enrichment is needed, would it be worth to suggest that more investigations in developing a protocol for enriching animals (depending on the study species) in laboratory is needed? That is to say instead of having a vague and general guidelines of ‘provide enrichment for animals’, it would be preferred to have something like ‘three or more times per week with novel objects’.
Others:
Without line number, it is a bit difficult to directly pinpoint the issue. I have tried to use page number, the number of line and section title to locate each point, hope it is clear:
1) P.2 line 4 under section ‘Environmental enrichment and drugs of abuse’. Sentence ‘I has been hypothesized that repeated novelty exposure’ - Replace ‘I’ by ‘It’
2) P. 2 under section ‘Environmental enrichment and drugs of abuse’ paragraph starts with ‘Interestingly’. What does ‘genetically vulnerable’ mean?
3) Second paragraph of ‘Enrichment as a treatment for attention deficit hyperactivity disorder’. The sentence ‘there is evidence that environmental enrichment actually increases inhibitory control for these children’, okay, but how? The next sentence also does not explain the ‘how’.
4) Section ‘Enrichment and social interaction’, the sentence ‘This may result from that fact that…’ this does not make sense to me. The previous sentence was stating a group of rats living in an enriched environment, but this sentence appear to say that rats that live together in a normal housing or non-enriched environments do not work out dominance hierarchies. That probably is not true.
5) Section ‘Enrichment and learning’ line 3. 1 ‘mo.’ – does this mean one month or one moment?
6) Section ‘Environmental Enrichment and complex behaviour’. How do psychology folks define ‘complex learning’?
7) With the section of learning appears to only have shown the outcome, but not the how – would there be a relationship between enrichment, neurochemical and learning?
8) Section ‘The suboptimal choice task.’ first paragraph. Need some references here, may be studies that used delay gratification task?
Author Response
Responses appear below each comment preceded by an *
Thank you for inviting me for reviewing this paper - It was an enjoyable read! The structure of the review is neat. The author shows the support for an enriched environment for animals. The view of the author is supported by studies that have shown the effects of enrichment on brain and behaviour. It covers a wide range of investigation topics. I entirely agree with the author that voices are increasing toward providing better welfare for animals. This review comes in timely and it will be a great start for the community to push more investigations forward, and hopefully come up a standard and basic protocol for keeping animals in the lab.
I only have a couple of minor comments, mostly about clarity (for people who are new to animal welfare, psychology etc but interested in these fields).
The author has only put forward the investigations in enrichment and complex behaviour (timing). But indeed, there are many sections missing the linkage between how enrichment affects learning performance, decrease impulsivity and increase attention for humans who have ADHD etc, would it be worth to have a section to talk about this relationship (i.e., a relationship between enrichment, neurochemical and learning)? Alternatively, the author could stress on the aim is to show the outcome (and not the 'how' specifically) of enrichment has on the brain and behaviour? I understand there is word limit here, but adding either of this will help to minimise queries (about the 'how) and focus on the main aim of the review.
I have added a sentence noting that the mechanisms for the effects of enrichment are not well understood.
Currently, most labs have their own way of giving enrichment to their animals, so as the author mentioned at the beginning that protocol in studies about enrichment varied. If the author strongly supports enrichment is needed, would it be worth to suggest that more investigations in developing a protocol for enriching animals (depending on the study species) in laboratory is needed? That is to say instead of having a vague and general guidelines of ‘provide enrichment for animals’, it would be preferred to have something like ‘three or more times per week with novel objects’.
*on p. 1, I have added a sentence noting that the nature of the enrichment needs to be further studied.
Others:
Without line number, it is a bit difficult to directly pinpoint the issue. I have tried to use page number, the number of line and section title to locate each point, hope it is clear:
1) P.2 line 4 under section ‘Environmental enrichment and drugs of abuse’. Sentence ‘I has been hypothesized that repeated novelty exposure’ - Replace ‘I’ by ‘It’
*fixed
2) P. 2 under section ‘Environmental enrichment and drugs of abuse’ paragraph starts with ‘Interestingly’. What does ‘genetically vulnerable’ mean?
*the authors used that term but I don’t think it is clear that it is genetic so we deleted the term.
3) Second paragraph of ‘Enrichment as a treatment for attention deficit hyperactivity disorder’. The sentence ‘there is evidence that environmental enrichment actually increases inhibitory control for these children’, okay, but how? The next sentence also does not explain the ‘how’.
*I have now explained how that might work.
4) Section ‘Enrichment and social interaction’, the sentence ‘This may result from that fact that…’ this does not make sense to me. The previous sentence was stating a group of rats living in an enriched environment, but this sentence appear to say that rats that live together in a normal housing or non-enriched environments do not work out dominance hierarchies. That probably is not true.
*the word “that” should be “the”
5) Section ‘Enrichment and learning’ line 3. 1 ‘mo.’ – does this mean one month or one moment?
*month
6) Section ‘Environmental Enrichment and complex behaviour’. How do psychology folks define ‘complex learning’?
*now defined
7) With the section of learning appears to only have shown the outcome, but not the how – would there be a relationship between enrichment, neurochemical and learning?
*I am not sure, but I have added a sentence at the end of the section on Enrichment and learning with speculation.
8) Section ‘The suboptimal choice task.’ first paragraph. Need some references here, may be studies that used delay gratification task?
*refs added.